# Study of Lipophilicity and ADME Properties of 1,9-Diazaphenothiazines with Anticancer Action

**DOI:** 10.3390/ijms24086970

**Published:** 2023-04-09

**Authors:** Beata Morak-Młodawska, Małgorzata Jeleń, Emilia Martula, Rafał Korlacki

**Affiliations:** 1Department of Organic Chemistry, Faculty of Pharmaceutical Sciences in Sosnowiec, The Medical University of SilesiaJagiellońska 4, 41-200 Sosnowiec, Poland; 2Doctoral School, The Medical University of Silesia, 40-055 Katowice, Poland; 3Department of Electrical and Computer Engineering, University of Nebraska-Lincoln, Lincoln, NE 68588, USA

**Keywords:** lipophilicity, 1,9-diazaphenothiazines, anticancer action, ADME, Lipinski’s rule of five, Ghose’s and Veber’s rules

## Abstract

Lipophilicity is one of the key properties of a potential drug that determines the solubility, the ability to penetrate through cell barriers, and transport to the molecular target. It affects pharmacokinetic processes such as adsorption, distribution, metabolism, excretion (ADME). The 10-substituted 1,9-diazaphenothiazines show promising if not impressive in vitro anticancer potential, which is associated with the activation of the mitochondrial apoptosis pathway connected with to induction BAX, forming a channel in MOMP and releasing cytochrome c for the activation of caspases 9 and 3. In this publication, the lipophilicity of previously obtained 1,9-diazaphenothiazines was determined theoretically using various computer programs and experimentally using reverse-phase thin-layer chromatography (RP-TLC) and a standard curve. The study presents other physicochemical, pharmacokinetic, and toxicological properties affecting the bioavailability of the test compounds. ADME analysis was determined in silico using the SwissADME server. Molecular targets studies were identified in silico using the SwissTargetPrediction server. Lipinski’s rule of five, Ghose’s, and Veber’s rules were checked for the tested compounds, confirming their bioavailability.

## 1. Introduction

Lipophilicity is one of the important physicochemical descriptors and is related to both pharmacodynamic and pharmacokinetic properties. It determines the absorption, metabolism, distribution, excretion and toxicity of drugs (ADMET) [1,2,3,4]. It plays a significant role in the transport of molecules across membranes. In addition, it affects their ability to bind to plasma proteins and to bind to receptors at the target of the drug’s action. Therefore, lipophilicity is regarded as the reference parameter for predicting the biological activity of potential drugs. Physically, lipophilicity is described as the logarithmic n-octanol-water partition coefficient (*logP*) that is characteristic of a given chemical. This parameter has been used in studies on the quantitative relationship between the structure and the activity (QSAR) [5,6,7].

According to Lipinski’s rule of five, lipophilicity is one of the most important factors determining the bioavailability of a drug. These are criteria that a chemical must meet in order to be considered a likely oral drug [8,9,10]. It has been proven that the *logP* > 5 lipophilicity value is associated with undesirable features of a given drug, such as tissue accumulation, fast metabolic turnover, poor water solubility, or strong plasma protein binding [11]. There is an exact correlation between lipophilicity and the permeability and solubility of a bioactive compound. In 2003, a published study showed the relationship between skin permeability and lipophilicity. It was also shown that too high values of lipophilicity contribute to the immobilization of the compound within a given layer [12]. Lipophilicity is inevitably associated with the penetration problem of blood–brain barrier (BBB). High lipophilicity promotes the nonspecific binding of drug molecules with plasma proteins, thus contributing to the reduced penetration of these compounds through the BBB. The literature data indicate that compounds with a moderate lipophilicity oscillating around 2 show optimal abilities to reach molecular targets [13,14,15].

Recently, we described new 10-substituted-1,9-diazaphenothiazines possessing promising anticancer activity. These compounds were obtained in organic synthesis using microwave radiation [16]. Such syntheses are currently highly efficient and selective [17,18]. These derivatives having in their structure various alkyl, alkynyl, cycloalkylaminoalkyl and dialkylaminoalkyl substituents were screened for their anticancer activity against using the glioblastoma SNB-19, melanoma C-32, and breast cancer MDA-MB-231 cell lines [16,19].

The parent compound 10*H*-1,9-diazaphenothiazine **1** was very active against melanoma C-32 (IC_50_ = 3.83 µM), even more potent than cisplatin (IC_50_ = 13.2 µM), but inactive against other lines. The most promising derivatives in this group were compounds **4** and **7** with the propynyl and diethylaminoethyl groups (Figure 1).

For those two compounds, the expression of *H3*, *TP53*, *CDKN1A*, *BCL-2,* and *BAX* genes was detected by the RT-QPCR method showing the induction of mitochondrial apoptosis. The results of gene expression analysis indicated that compounds **4** and **7** selectively reduced the expression of the *H3* gene, and as well the expression of the *TP53* gene, but enhanced the expression of the *CDKN1A* gene in the tested cell lines. The induction was confirmed in the BAX/BCL-2 gene expression ratio studies of mitochondrial apoptosis in two cancer cell lines (SNB-19 and MDA-MB-231). In the C-32 melanoma cell line, a transcription gene activity indicated a different mode of cell death. The Annexin V apoptosis detection assay showed populations corresponding to viable, necrotic, early and late apoptopic cells. While SNB-19 cells were treated with compounds **4** and **7**, there was a slight increase in the early and late cell apoptosis populations and a slight decrease in the viable cell population. In order to thoroughly understand the mechanism of action of derivatives **4** and **7**, the determination of proteins related to apoptosis was performed using the Proteome Profiler Human Apoptosis Array. Twelve expressed proteins were confirmed in the study. We found the following proteins involved in apoptosis: BAX, pro-caspase-3, cytochrome c, and SMAC/Diablo. Compounds **4** and **7** are indicated to be highly likely to induce BAX, as they form a channel in MOMP and release cytochrome c for the activation of caspases 9 and 3 thereby initiating apoptosis via the intrinsic mitochondrial pathway [19]. These results prompted further research in the field of 1,9-diazaphenothiazines, including pharmacokinetic analyses, in particular lipophilicity and the target analysis.

The purpose of this work is to determine the lipophilicity parameters (*logP_calcd_*, *R_M_*_0_ and *logP_TLC_*) of eleven new anticancer 10-substituted 1,9-diazaphenothiazines **1**–**11** by computational programs and by the RP-TLC method; to discuss the influence of the nature of the substituents; to compare the calculated data with the experimental ones; and to analyze of the molecular descriptors and ADME properties.

## 2. Results

The research began with the calculation of lipophilicity parameters with the use of eleven popular computer modules available on the VCCLAB server [20] and SwissADME [21] server. These programs are based on various mathematical algorithms. The calculated *logP_calcd_* values for the 10-substituted 1,9-diazaphenothiazines differed depending on the substituents on the thiazine nitrogen atom and on the calculation program (Table 1). The 10*H*-1,9-diazaphenothiazine **1** was characterized by the lowest lipophilicity (*logP_calc_*_d_ = 1.51), which was obtained by the MlogP program. The highest lipophilicity was obtained for the 10-benzyl-1,9-diazaphenothiazine **5** (*logP_calcd_* = 4.75) according to the XLOGP2 module. In contrast, derivatives with the dialkylaminoalkyl or cycloalkylamino substituents **6**–**11** were characterized by lipophilicity in the range from 2.02 to 3.89.

In the next stage, experimental measurements of the lipophilicity coefficient were performed with the use of reversed-phase thin-layer chromatography RP-TLC. First, the relative lipophilicity of these derivatives **1**–**11** was measured as expressed by the *R_M_*_0_ chromatographic values.

The experimental RP-TLC method provided the *R_M_* retention parameter (calculated from R_F_ value measurements) using the following equation:*R_M_* = *log* (1/R_F_ − 1)(1)

The *R_M_* values decreased linearly with increasing acetone concentration in the mobile phase (*r* = 0.9485–0.9985). Extrapolation to zero acetone concentration provided the values of the relative lipophilicity parameter (*R_M_*_0_), which showed the division between the non-polar stationary phase and the polar mobile phase, according to the equation:*R_M_* = *R_M_*_0_ + *b*C,(2)
where C is the concentration of acetone. The *R_M_*_0_ values of 10-substituted 1,9-diazaphenothiazines are in the range 1.1970–1.9988 (Table 2).

Then a calibration curve was created using the same measuring conditions. A set of reference substances **I**–**V** with the literature values of *logP_lit._* was used in the range from 1.21 to 3.54 (Table 3). The calibration curve was used to convert the values of the relative lipophilicity parameter *R_M_*_0_ of the tested hybrids into the value of the absolute lipophilicity parameter *logP_TLC_*.

The equation of the standard curve with which it was possible to convert the relative lipophilicity parameter *R_M_*_0_ into the absolute parameter *logP_TLC_* is presented in Figure 2.

The *logP_TLC_* values for all new anticancer 10-substituted 1,9-diazphenothiazines **1**–**11** are collected in Table 4. 

Simultaneously with conducting the experimental studies, the analyzes of molecular descriptors, Lipinski’s, Ghose’s, and Veber’s parameters were performed using the SwissADME server (Table 5) [21]. 

ADME parameters were performed using the PreADMET server (Table 6) [24].

As shown in Table 5 and Table 6, the compounds studied show significant differences in molecular descriptors as well as their ADME parameters. All tested derivatives, however, meet the requirements of Lipinski’s five rule, and Ghose’s, and Veber’s rules [21] (Table 5).

The values of the relative lipophilicity parameter *R_M_*_0_ of the tested molecules **1**–**11** were correlated with the ADME parameters determined in silico. The results are presented in Table 7.

An analysis of the relationship between the lipophilicity and the polarity of the studied molecules **1**–**11** was also performed using the BOILED-Egg method of estimating the penetration through the brain or intestines, as an accurate predictive model [21,25]. This analysis is presented in Figure 3. The tested compounds are within the range of good permeability through the blood–brain barrier (BBB) (yellow area) and the proper binding to the human blood albumin (HIA) (white area). Additionally, compounds that can become substrates for p-glycoprotein (blue points) and two derivatives **3**, **8** that show a negative result (red points) have been indicated in the Figure 3.

Using SwissTargetPrediction server [26], the molecular targets that are likely to be achieved by reached tested 1,9-diazphenothiazines were also predicted (Table 8). These include the family CG and AG protein, kinase, protease, and cytochrome 450.

## 3. Discussion

The attention of this work was focused on the evaluation of lipophilicity, physicochemical properties, and molecular targets of new anticancer-active 10-substituted 1,9-diazaphenothiazines **1**–**11**, which have different substituents at the thiazine nitrogen atom: the alkyl, alkenyl, alkynyl, dialkylaminoalkyl, and cycloalkylaminoalkyl. The synthesis, the structure, and high biological potential of these derivatives have been previously documented [19]. These compounds showed significant and highly promising anticancer activity as determined in vitro against the glioblastoma SNB-19, the melanoma C-32, and the breast cancer MDA-MB-231 cell lines. The most active, 1,9-diazaphenothiazines **4** and **7**, were analyzed in terms of the expression of genes affecting the neoplastic process (*H3*, *TP53*, *CDKN1A*, *BCL-2*, and *BAX*). These studies showed the activation of the mitochondrial apoptotic pathway and the destruction of normal histone formation.

Lipophilic studies were started with analyzes in silico using available VCCLAB and SwissADME web servers. These studies use various mathematical modules described on the websites of the above servers. The obtained results of the calculated lipophilicity fall into a wide range of values. This is most likely the result of the use of different computational models. The most lipophilic compound was derivative **5** (*logP_calcd_* = 4.75) with a benzyl substituent. The least lipophilic compounds were native 10*H*-1,9-diazaphenothiazine **1** (*logP_calcd_* = 1.51) with a small hydrogen atom at position 10. The results of these studies are included in Table 1, and the graphical visualization of the calculated *logP* values of each compound is shown in Figure 3 and Figure 4. Such large differences in the obtained values of the lipophilicity parameter were reported before [27,28,29] and, as in these previous studies, accurate experimental measurments are needed in order to narrow them down.

Experimental studies were performed to determine the relative lipophilicity parameter *R_M_*_0_ according to the methodology described in Section 2 and Section 4. The tested derivatives are characterized by rather lower (below 2) *R_M_*_0_ parameters, which are in the range from 1.197 to 1.9781. The drug prothipendyl **12**, which is a representative of 1-azaphenothiazines, was used as a reference substance in the tests. It can be seen that the presence of an additional second nitrogen atom in the phenothiazine structure substantially reduces the lipophilicity of these derivatives.

Subsequently, a calibration curve was made to determine the absolute lipophilicity parameter *logP*. In this process, the following reference substances with a known lipophilicity parameter *logP* were used: acetanilide **I**, acetophenone **II**, 4-bromoacetophenone **III**, benzophenone **IV**, anthracene **V**, for which the literature *logP_lit_* values ranged from 1.21 to 5.53 (Table 3) [22,23].

Using the calibration curve equation (Figure 2), the relative lipophilicity parameter *R_M_*_0_ was converted to the absolute values *logP*, which are summarized in Table 4. They are in the range from 1.476 to 2.247 (Table 4). The lowest parameter is characterized by 10*H*-1,9-diazaphenothiazine **1**, and the highest parameter has the derivative **11** containing a cycloaminoalkyl substituent. However, these values differ from the computer-calculated parameters, which is also shown in Figure 4. The results obtained depend on the type of substituent on the thiazine nitrogen atom. When analyzing the lipophilicity in terms of anticancer activity, it should be noted that the most anticancer-active compounds **4** and **7** have a similar lipophilicity value with the other derivatives, which is in the range from 1.4 to 2.3. This is, therefore, an indication that the lipophilicity parameter is only one of many factors affecting the biological potential.

It can be concluded that the group of 10-substituted 1,9-diazaphenothiazines is moderately lipophilic compared to the isomeric 1,6-, 1,8-, 2,7 and 3,6-diazaphenothiazines [29,30,31,32,33]. A comparative summary of selected derivatives with the following substituents, hydrogen atom (**A** derivatives), allyl (**B** derivatives), propargyl (**C** derivatives) and dimethylaminopropyl (**D** derivatives), is presented in Table 9 and visualized graphically in Figure 5.

Among the 10*H*-dipyridothiazines (derivatives **A**) shown as the dark blue curve (Figure 5), 10*H*-3,6-diazaphenothiazine is the least lipophilic and 10*H*-1,6-diazaphenothiazine is the most lipophilic. A similar relationship exists for derivatives with an allyl substituent (**B**) (red curve). The least lipophilic dipyridothiazine with a propargyl substituent is 3,6-diazaphenothiazine and the most lipophilic is 1,8-diazaphenothiazine, as shown by the green curve. Among the derivatives with a dimethylaminopropyl substituent (derivatives **D**), the lowest lipophilicity is characterized by 1,6-diazaphenothiazine and the highest by 1,9-diazaphenothiazine (yellow curve). It is also a proof that lipophilicity depends on both the diazaphenothiazine system and the type of substituent at the thiazine nitrogen atom. 

The molecular descriptors of all 1,9-diazaphenothiazines were thoroughly analyzed against the requirements of the five rules of Lipinski, and the rules of Ghose and Veber (Table 5). All tested derivatives meet the requirements of the Lipinski rule of five, and of the Ghose and Veber rules. These results indicate that the tested 1,9-diazaphenothiazine derivatives can become drugs and specifically orally active drugs.

ADME analyzes were performed for the tested compounds using PreADMET server [24]. The results obtained in comparison to the reference compound, prothipendyl **12**, turn out to be quite interesting (Table 6). The tested compounds have the blood–brain barrier penetration index BBB in the range from 0.85 to 4.55, for most of them significantly different from that of the reference compound **12** (2.40). The differences in blood–brain barrier penetration appear to depend on the structure of the tested derivative. Caco-2 cell permeability was different among the tested compounds. All tested compounds exhibited a high HIA index, which was in the range from 95 to 98. The permeability of MDCK cells was variable and ranged from 3.73 to 80.31. These values differ significantly in the studied group of compounds and depend on the type of substituent in the 1,9-diazaphenothiazine system. Similar differences were observed for the PPB parameter. All test compounds have a very similar and substantially low SP parameter compared to the reference compound. 

The ADME parameters (BBB, Caco-2, HIA, MDCK, PPB, SP) determined in silico were correlated with the relative lipophilicity parameter *R_M_*_0_ (Table 8). Correlations were obtained as polynomial equations of the second and third degree with the *r* values in the range from 0.6205 to 0.8616. The obtained results are an indication that lipophilicity is one of many factors that can directly affect the biological activity. The obtained correlations also show that lipophilicity is dependent on the conformation of molecules, hydrogen bonds, ionic interactions as well as interactions related to van der Waals forces. 

Using the SwissADME server [21], we performed an analysis of the relationship between the lipophilicity of the tested 1,9-diazaphenothiazines **1**–**11** and their polarity. The BOILED-Egg method was used, which estimates permeation through the brain and intestines. This analysis is shown in Figure 3. All tested compounds fall within the range of good permeability across the blood–brain barrier (BBB) as indicated by the yellow region and normal binding to human blood albumin (HIA) by the white region. In addition, derivatives **1**, **2**, **4**–**7**, **9**–**11** that can become p-glycoprotein substrates (blue points) and two derivatives **3, 8** that give a negative result (red points) are indicated.

At the latest stage, the determination of molecular targets was performed using the SwissTargetPrediction server (Table 8) in order to confirm their anticancer potential. The group of 10-substituted 1,9-diazaphenothiazines may affect the activity of the family CG and AG protein, kinase, protease, and cytochrome 450. The obtained results are promising and are expected to inspire further research at the in vitro level.

## 4. Materials and Methods

### 4.1. Reagents

10-Substituted 1,9-diazaphenothiazines **1**–**11** were obtained in the reactions described recently [16]. The reference compound prothipendyl (chemical name: 10-dimethylaminopropyl-1-azaphenothiazine) **12** (AWD Pharma Germany) was used.

The following reagents were used in the experimental studies to prepare the mobile phase: acetone, (POCh, Gliwice, Poland), TRIS (tris (hydroxymethyl) aminomethane, Fluka). In order to prepare the calibration curve, five chemical compounds with the described lipophilicity parameter (log*P_lit_*.) were used: acetanilide (POCh, Gliwice, Poland), acetophenone (POCh, Gliwice, Poland), 4-bromoacetophenone (Fluka, Buchs, Switzerland), benzophenone (Fluka, Buchs, Switzerland), antracene (POCh, Gliwice, Poland). 

### 4.2. Chromatographic Procedure

The RP-TLC method was used to determine the experimental lipophilicity according to the literature [30,31,32,33]. Modified silica gel RP 18F_254S_ (Merck) was used as the stationary phase and a mixture of tris(hydroxymethyl)aminomethane (TRIS) (0.2 M, buffer pH = 7.4) with acetone as the mobile phase with the range of concentrations from 40 to 70% (*v*/*v*) in 5% increments. The compounds **1**–**12** and the standards **I-V** were dissolved in ethanol (2.0 mg/mL) and 2 μL of each solution was spotted. Spots were observed under UV light at λ = 254 nm. Each measurement was performed in triplicate and then the R_F_ values were calculated.

### 4.3. Theoretical Lipophilicity and ADMET Parameters, Target Prediction

The calculated lipophilicity was determined using various internet servers: VCCLAB [20] and SwissADME [21] including: Alogps, AC_Logp, ALOGP, MLOGP, XLOGP2, XLOGP3, ILopP, XlogP, WlogP, MlogP, SILICOS-IT. The molecular descriptor and ADME parameters were calculated using SwissADME and PreADME server [21,24]. Target prediction was determined by the SwissTargetPrediction server [26].

## 5. Conclusions

The presented research results show the lipophilicity of the group of 10-substituted 1,9-diazaphenothiazines showing high anticancer potential, associated with the disruption of the function of the p53 protein and the proper functioning of histones, which was documented in previous studies. Lipophilicity as a parameter determining the reaching of the molecules to the biological target was determined theoretically by computational methods, and experimentally using reverse-phase thin-layer chromatography (RP-TLC).

The compounds tested were moderately lipophilic compared to the previously reported 1,6-, 1,8-, 2,7-, and 3,6-diazaphenothiazine derivatives with analogous substituents. In addition, ADME parameters that are indisputably related to lipophilicity have been determined. The new derivatives followed the rules of Lipinski, Ghose, and Veber, indicating that they may become oral drugs in the future. Further studies of this group of compounds in order to fully determine the pharmacological potential of these derivatives have been planned.

## Figures and Tables

**Figure 1 ijms-24-06970-f001:**
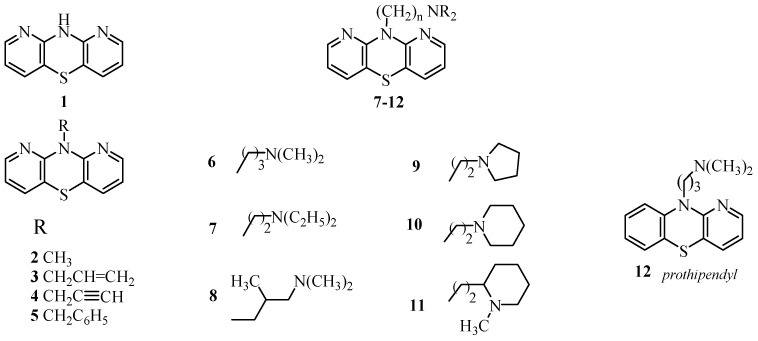
Structure of novel 10*H*-1,9-diazaphenothiazine **1** and 10-substituted 1,9-diazaphenothiazines **2**–**11** and reference compound prothipendyl **12**.

**Figure 2 ijms-24-06970-f002:**
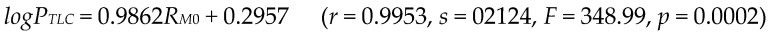
The standard curve equation prepared with standard substances **I**–**V**.

**Figure 3 ijms-24-06970-f003:**
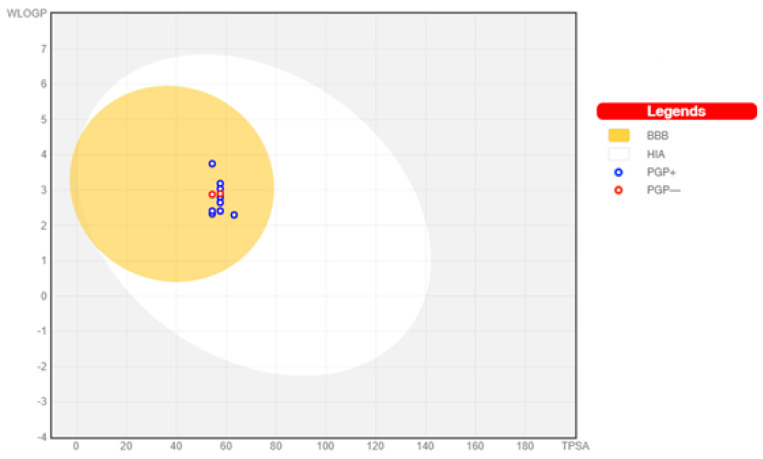
Graph of the dependence of lipophilicity on the polarity of the studied molecules **1**–**11**, determined by the BOILED-Egg method [21].

**Figure 4 ijms-24-06970-f004:**
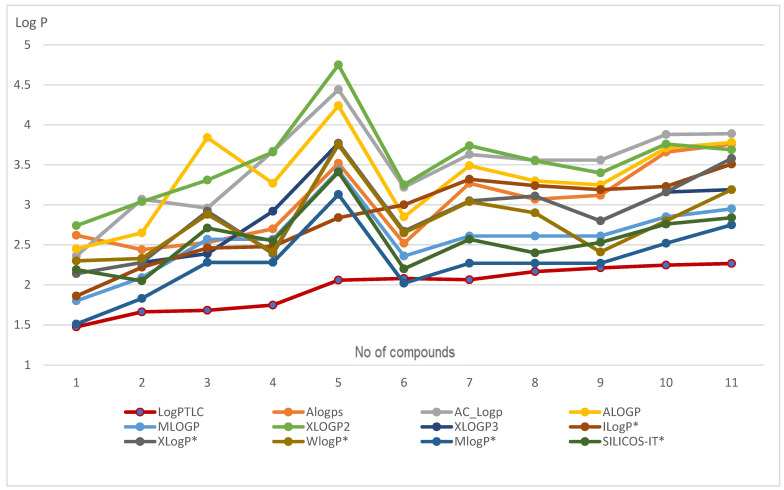
Graphical visualization of calculated *logP* values (using SwissADME models) of the tested compounds with comparison of *logP_TLC_* (plotted in red). *-data obtained using the SwissADME server.

**Figure 5 ijms-24-06970-f005:**
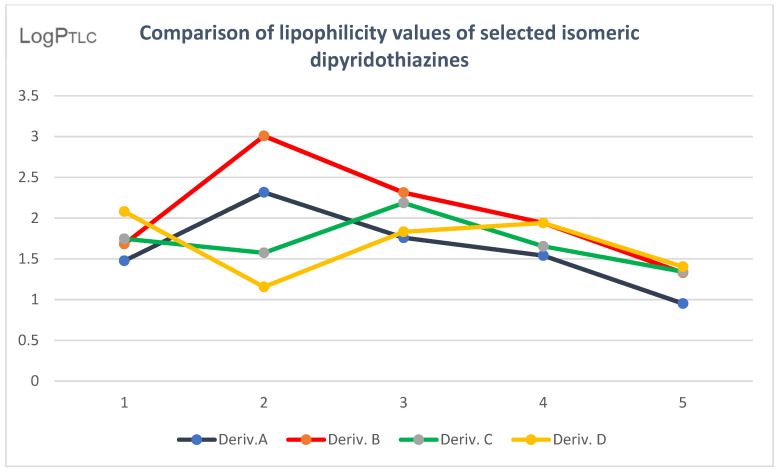
Graphical visualization of log*P_TLC_* values of the selected derivatives (**A**–**D**) of 1,9- 1,6-, 1,8-, 2,7- and 3,6-diazaphenothiazines [30,31,32,33].

**Table 1 ijms-24-06970-t001:** The calculated lipophilic parameters (*logP_calcd_*) for 1,9-dipyridothiazine **1**–**10** using the internet data bases: VCCLAB [20] and SwissADME * [21].

No	Alogps	AC_Logp	ALOGP	MLOGP	XLOGP2	XLOGP3	ILogP *	XLogP *	WlogP *	MlogP *	SILICOS-IT *
**1**	2.62	2.36	2.45	1.80	2.74	2.14	1.86	2.14	2.30	1.51	2.19
**2**	2.44	3.07	2.65	2.09	3.04	2.28	2.22	2.28	2.33	1.83	2.05
**3**	2.53	2.96	3.84	2.57	3.31	2.39	2.46	2.92	2.88	2.28	2.71
**4**	2.70	3.67	3.27	2.57	3.66	2.92	2.48	2.39	2.41	2.28	2.55
**5**	3.52	4.44	4.24	3.43	4.75	3.77	2.84	3.77	3.75	3.13	3.41
**6**	2.52	3.22	2.85	2.36	3.25	2.67	3.00	2.67	2.65	2.02	2.20
**7**	3.27	3.63	3.49	2.61	3.74	3.05	3.32	3.05	3.04	2.27	2.57
**8**	3.07	3.56	3.30	2.61	3.55	3.11	3.24	3.11	2.90	2.27	2.40
**9**	3.12	3.56	3.25	2.61	3.40	2.80	3.19	2.80	2.41	2.27	2.53
**10**	3.66	3.88	3.71	2.85	3.76	3.16	3.23	3.16	2.80	2.52	2.76
**11**	3.76	3.89	3.78	2.95	3.69	3.19	3.51	3.58	3.19	2.75	2.84

*-data obtained using the SwissADME server.

**Table 2 ijms-24-06970-t002:** The *R_M_*_0_ values and *b* (slope) and *r* (correlation coefficient) of the equation *R_M_* = *R_M_*_0_ + *b*C for compounds **1**–**11**.

No	*−b*	*R_M_* _0_	*r*
**1**	0.0205	1.1970	0.9916
**2**	0.0192	1.3850	0.9954
**3**	0.0278	1.4052	0.9949
**4**	0.0195	1.4720	0.9985
**5**	0.0205	1.7877	0.9868
**6**	0.0170	1.8096	0.9973
**7**	0.0180	1.7934	0.9168
**8**	0.0228	1.8961	0.9319
**9**	0.0123	1.9434	0.9891
**10**	0.0214	1.9781	0.9485
**11**	0.0211	1.9988	0.9686

**Table 3 ijms-24-06970-t003:** *R*_*M*0_ and *logP_lit_*. values and *b* (slope) and *r* (correlation coefficient) of the equation *R*_*M*_ = *R*_*M*0_ + *b*C for standards **I**–**V**.

Parameters	I	II	III	IV	V
*log* *P* _TLC_	1.21 [22]	1.58 [22]	2.43 [23]	3.18 [22]	5.53 [22]
*R* _*M*0_	1.011	1.601	2.281	2.996	3.588
−*b*	0.018	0.019	0.033	0.034	0.044
*r*	0.9964	0.9967	0.9961	0.9842	0.9864

**Table 4 ijms-24-06970-t004:** The *logP_TLC_* values of investigated compounds **1**–**11**.

	No of Investigated Compounds										
	1	2	3	4	5	6	7	8	9	10	11
*logP* * _TLC_ *	1.476	1.662	1.682	1.747	2.059	2.080	2.064	2.166	2.212	2.247	2.267

**Table 5 ijms-24-06970-t005:** The molecular descriptor and parameters of Lipinski’s, Ghose’s, and Veber’s rules for 1,9-dipyridothiazines **1**–**11** and prothipendyl **12 [21]**.

No	Molecular Mass (M)	H-Bond Acceptors	H-Bond Donors	Rotatable Bonds	Molar Refractivity	TPSA	P-gp Substrate	Lipinski’s Rules	Ghose’s Rules	Veber’s Rules
**1**	201	2	1	0	58.63	63.11	+	+	+	+
**2**	215	2	0	0	63.54	54.32	+	+	+	+
**3**	241	2	0	2	72.68	54.32	-	+	+	+
**4**	239	2	0	1	71.31	54.32	+	+	+	+
**5**	291	2	0	2	88.02	54.32	+	+	+	+
**6**	286	3	0	4	85.66	57.56	+	+	+	+
**7**	300	3	0	5	90.47	57.56	+	+	+	+
**8**	300	3	0	4	90.47	57.56	-	+	+	+
**9**	298	3	0	3	92.27	57.56	+	+	+	+
**10**	312	3	0	3	97.07	57.07	+	+	+	+
**11**	326	3	0	3	101.88	57.56	+	+	+	+

**Table 6 ijms-24-06970-t006:** The ADME activities predicted for 1,9-dipyridothiazines **1**–**11** and prothipendyl **12 [24]**.

No	1	2	3	4	5	6	7	8	9	10	11	12
BBB	1.02	2.36	3.06	3.36	4.45	2.40	3.17	2.97	2.91	1.26	0.85	2.40
Caco-2	25.41	27.81	25.07	24.30	25.39	22.55	22.61	22.81	24.14	24.00	23.92	22.55
HIA	95.81	98.05	97.67	97.71	97.37	97.78	97.66	97.66	97.66	97.57	97.49	97.78
MDCK	59.28	52.80	26.47	9.12	3.73	37.63	80.31	26.77	60.18	52.03	9.17	37.63
PPB	99.77	93.89	88.37	88.12	96.53	68.57	78.56	74.82	74.86	82.47	80.83	68.57
SP	−3.36	−3.24	−3.14	−2.92	−2.94	−3.50	−3.32	−3.36	−3.76	−3.62	−3.58	−3.50

**Table 7 ijms-24-06970-t007:** The correlation of the *R_M_*_0_ values with the predicted ADME parameters.

No of Compound	ADME Activities	Equation	*r*
**1–11**	BBB	*BBB* = 14.86 *R_M_*_0_^3^ + 54.304 *R_M_*_0_^2^ − 57.146 *R_M_*_0_ + 17.105	0.8616
**1–11**	Caco-2	*Caco-*2 = 37.616 *R_M_*_0_^3^ − 178.2 *R_M_*_0_^2^ + 273.43 *R_M_*_0_ − 110.91	0.7191
**1–11**	HIA	*HIA* = 7.29 *R_M_*_0_^2^ + 24.557 *R_M_*_0_ + 77.334	0.7962
**1–11**	MDCK	*MDCK* = 529.07 *R_M_*_0_^3^ + 2648.5 *R_M_*_0_^2^ + 93449 *R_M_*_0_ – 35,341	0.6205
**1–11**	PPB	*PPB* = 30.856 *R_M_*_0_^2^ − 125.38 *R_M_*_0_ + 205.96	0.7335
**1–11**	SP	*SP* = −3.0033 *R_M_*_0_^2^ + 9.2228 *R_M_*_0_ − 10.128	0.8115

**Table 8 ijms-24-06970-t008:** Probable target classes of 1,9-dipyridothiazines **1**–**11** [26].

No	Probable Targets
**1**	Family C G protein, unclasificated protein, enzyme
**2**	Proteaze, Family C G protein, cytochrome 450
**3**	Cytochrome 450, kinase, electrochemical transporter
**4**	Unclasificated protein, enzyme, kinase
**5**	Phosphodiesterase, enzyme, kinase
**6**	Kinase, protease, family AG protein
**7**	Enzyme, protease, family AG protein
**8**	Family AG protein, kinase, ligand-gate ion channel
**9**	Family AG protein, ligand-gate ion channel, kinase
**10**	Family AG protein, ligand-gate ion channel, kinase
**11**	Family AG protein, ligand-gate ion channel, kinase

**Table 9 ijms-24-06970-t009:** Comparison of the lipophilicity parameter *logP_TLC_* of selected isomeric diazaphenothiazines.

SubstituentR	1,9-Diaza 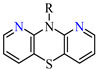	1,6-Diaza [30] 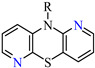	1,8-Diaza [31] 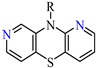	2,7-Diaza [32,33] 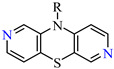	3,6-Diaza [30] 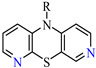
A. H	1.476	2.316	1.759	1.540	0.952
B. CH_2_CH=CH_2_	1.682	3.008	2.313	1.940	1.329
C. CH_2_CCH	1.747	1.574	2.186	1.654	1.342
D. CH_2_CH_2_CH_2_N(CH_3_)_2_	2.080	1.156	1.832	1.940	1.404

## Data Availability

Not applicable.

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
