# Peer review of "Study of Lipophilicity and ADME Properties of 1,9-Diazaphenothiazines with Anticancer Action"

_ijms, 2023, doi:10.3390/ijms24086970_

Round 1

Reviewer 1 Report

This work was focused on the evaluation of lipophilicity, physicochemical properties, and molecular targets 10-substituted 1,9-diazaphenothiazines 1-11 by various computer programs.

Firstly, it’s better to clearly state the significance of the article, purely predicting its parameters is unattractive for a research paper.

The author predicts ADME by computer. Is there any difference between the predicted result and the actual result?

The author said that lipophilicity has a great relationship with ADME, which is undoubtedly true, however, the author should analyze the results of lipophilicity and ADME in this study to show the relationship between them.

 The error bar was lost in Fig5.

Author Response

The response to the reviewer 1

Thank you for all comments and suggestions:

This work was focused on the evaluation of lipophilicity, physicochemical properties, and molecular targets 10-substituted 1,9-diazaphenothiazines 1-11 by various computer programs.

1.Firstly, it’s better to clearly state the significance of the article, purely predicting its parameters is unattractive for a research paper.

Answer: The purpose of the prepared manuscript was written clear in the prepared abstract. The conducted research concerns a group of compounds with high anti-cancer potential and includes studies of lipophilicity in experimental and computational terms as well as ADME parameters in order to determine the pharmacokinetics of these derivatives. Similar publications on this topic have been published previously e.g.:

  1. Kadela-Tomanek, M.; Jastrzębska, M; Chrobak, E.; Bębenek, E. Lipophilicity and ADMET Analysis of Quinoline-1,4-quinoneHybrids. Pharmaceutics 2023, 15, 34.
  2. Morak‐Młodawska, B.; Pluta, K.; Jeleń, M. Estimation of the Lipophilicity of New Anticancer and Immunosuppressive 1,8‐Diazaphenothiazine Derivatives. J. Chrom. Sci. 2015, 53, 462–466.
  3. Morak‐Młodawska, B.; Jeleń. M. Lipophilicity and Pharmacokinetic Properties of New Anticancer Dipyridothiazine with 1,2,3-Triazole Substituents. Molecules 2022, 27, 1253.

2.The author predicts ADME by computer. Is there any difference between the predicted result and the actual result?

Answer: The predicted ADME results were made with the participation of Swiss servers, however, it should be borne in mind that these are computational results. These results correlate well with the lipophilicity parameter. Detailed experimental studies of these ADME parameters are the subject of our next project.

  1. The author said that lipophilicity has a great relationship with ADME, which is undoubtedly true, however, the author should analyze the results of lipophilicity and ADME in this study to show the relationship between them.

Answer: The lipophilicity and ADME parameters obtained in our study were correlated and included in a new Table 7, and their relationship was discussed in the discussion.

  1. The error bar was lost in Fig5.

The form of the figure 5 has been changed to a scatter graph because it is a visualization of the data contained in Table 9.

In response to the review, we would like to thank you for evaluating our manuscript. All changes requested by other reviewers as well as linguistic corrections are marked in yellow.

Reviewer 2 Report

Manuscript can be accepted

Author Response

The response to the reviewer 2

Thank you for all comments and suggestions:

Comments and Suggestions for Authors: Manuscript can be accepted

Answer: In response to the review, we would like to thank you for evaluating our manuscript and for your favorable review. Changes requested by other reviewers as well as linguistic corrections are marked in yellow.

Reviewer 3 Report

The manuscript on title Innovation of Study of Lipophilicity and ADME Properties of 1,9-Diazaphe- 2 nothiazines with Anticancer Action It is suitable for publication after major revision.

[1] Title of manuscript should be more concise

[2]  Please improve the English language throughout the manuscript (spaces).

3] The letters "H" "N" in the name of compounds must be italic

4] Revise references according to Journal template

5] Add some recent related reference:

Molecules 27 (7), 2061, 2022

Current Organic Chemistry, Volume 26, Number 24, 2022, pp. 2214-2222(9)

ACS omega 7 (49), 45535-45544,2022

ACS Omega 7 (26), 22839–22849, 2022

Author Response

The response to the reviewer 3

Thank you for all comments and suggestions:

The manuscript on title Innovation of ”Study of Lipophilicity and ADME Properties of 1,9-Diazaphenothiazines with Anticancer Action”It is suitable for publication after major revision.

[1] Title of manuscript should be more concise.

Answer: We kindly ask you to keep the title, because it is similar to the titles of publications in the topic, e.g.:

  1. Kadela-Tomanek, M.; Jastrzębska, M; Chrobak, E.; Bębenek, E. Lipophilicity and ADMET Analysis of Quinoline-1,4-quinoneHybrids. Pharmaceutics 2023, 15, 34.
  2. Morak‐Młodawska, B.; Pluta, K.; Jeleń, M. Estimation of the Lipophilicity of New Anticancer and Immunosuppressive 1,8‐Diazaphenothiazine Derivatives. J. Chrom. Sci. 2015, 53, 462–466.
  3. Dołowy, M.; Jampilek, J.; Bober-Majnusz, K. A comparative study of the lipophilicity of metformin and phenformin. Molecules 2021, 26, 6613.

[2]  Please improve the English language throughout the manuscript (spaces).

Answer: A thorough correction of the English language has been made.

[3] The letters "H" "N" in the name of compounds must be italic

Answer: Compound names have been checked and corrected.

[4] Revise references according to Journal template

Answer: References have been corrected.

[5] Add some recent related reference:

Molecules 27 (7), 2061, 2022, Current Organic Chemistry, Volume 26, Number 24, 2022, pp. 2214-2222(9),ACS omega 7 (49), 45535-45544,2022, ACS Omega 7 (26), 22839–22849, 2022

Answer: References have been expanded and the indicated selected publications have been attached.

      In response to the review, we would like to thank you for evaluating our manuscript and for your favorable review. All changes requested by other reviewers as well as linguistic corrections are marked in yellow.

Reviewer 4 Report

Please include the error bars for the figure 5 when comparing the logP values. Apart from the rest of article is good to publish 

Author Response

The response to the reviewer 4

Thank you for all comments and suggestions.

Please include the error bars for the figure 5 when comparing the logP values. Apart from the rest of article is good to publish.

Answer:

The form of the figure 5 has been changed to a scatter graph because it is a visualization of the data contained in Table 9. In response to the review, we would like to thank you for evaluating our manuscript and for your favorable review. Changes requested by other reviewers as well as linguistic corrections are marked in yellow.

Round 2

Reviewer 1 Report

The author well reposed the raised questions. This revised manuscript can be published with this version.